# Gemfibrozil Improves Microcirculatory Oxygenation of Colon and Liver without Affecting Mitochondrial Function in a Model of Abdominal Sepsis in Rats

**DOI:** 10.3390/ijms25010262

**Published:** 2023-12-23

**Authors:** Anne Kuebart, Katharina Gross, Charlotte Maicher, Max Sonnenschein, Annika Raupach, Jan Schulz, Richard Truse, Stefan Hof, Carsten Marcus, Christian Vollmer, Inge Bauer, Olaf Picker, Borna Relja, Anna Herminghaus

**Affiliations:** 1Department of Anesthesiology, University Hospital Duesseldorf, Moorenstrasse 5, 40225 Duesseldorf, Germany; annekonstanzecharlotte.kuebart@med.uni-duesseldorf.de (A.K.);; 2Department of Trauma, Hand, Plastic and Reconstructive Surgery, Translational and Experimental Trauma Research, University Hospital Ulm, Ulm University, Albert-Einstein-Allee 23, 89081 Ulm, Germany

**Keywords:** sepsis, CASP, fibrates, gemfibrozil, microcirculation, mitochondrial function

## Abstract

Recent studies observed, despite an anti-hyperlipidaemic effect, a positive impact of fibrates on septic conditions. This study evaluates the effects of gemfibrozil on microcirculatory variables, mitochondrial function, and lipid peroxidation levels with regard to its potential role as an indicator for oxidative stress in the colon and liver under control and septic conditions and dependencies on PPARα-mediated mechanisms of action. With the approval of the local ethics committee, 120 Wistar rats were randomly divided into 12 groups. Sham and septic animals were treated with a vehicle, gemfibrozil (30 and 100 mg/kg BW), GW 6471 (1 mg/kg BW, PPARα inhibitor), or a combination of both drugs. Sepsis was induced via the colon ascendens stent peritonitis (CASP) model. Then, 24 h post sham or CASP surgery, a re-laparotomy was performed. Measures of vital parameters (heart rate (HR), mean arterial pressure (MAP), and microcirculation (µHbO_2_)) were recorded for 90 min. Mitochondrial respirometry and assessment of lipid peroxidation via a malondialdehyde (MDA) assay were performed on colon and liver tissues. In the untreated sham animals, microcirculation remained stable, while pre-treatment with gemfibrozil showed significant decreases in the microcirculatory oxygenation of the colon. In the CASP animals, µHbO_2_ levels in the colon and the liver were significantly decreased 90 min after laparotomy. Pre-treatment with gemfibrozil prevented the microcirculatory aberrations in both organs. Gemfibrozil did not affect mitochondrial function and lipid peroxidation levels in the sham or CASP animals. Gemfibrozil treatment influences microcirculation depending on the underlying condition. Gemfibrozil prevents sepsis-induced microcirculatory aberrances in the colon and liver PPARα-independently. In non-septic animals, gemfibrozil impairs the microcirculatory variables in the colon without affecting those in the liver.

## 1. Introduction

Sepsis remains an ongoing challenge in intensive care units worldwide. Once sepsis is diagnosed, mortality rates range around thirty percent, implying the need to improve sepsis management further [1]. 

Remarkably, next to other types of organ failure, liver dysfunction and intestinal dysfunction occur frequently in critically ill patients, both displaying independent risk factors of mortality [2,3,4]. Still, in contrast to other organ dysfunctions in cases of sepsis, long-term liver and intestinal dysfunction lack replacement methods or adequate therapeutic approaches. Therefore, it is even more crucial to prevent liver and intestinal damage by revealing the cause and underlying pathomechanisms and developing new therapeutic strategies. So far, it is known that impaired microcirculation displays one major hallmark in addressing these organ dysfunctions: it triggers microcirculatory tissue hypoxemia and organ damage and correlates with increased mortality [5]. Another hallmark might be mitochondrial dysfunction [6], provoking insufficient adenosine triphosphate (ATP) generation and leading to restricted cell functions, resulting in systemic organ dysfunctions [6]. Based on these findings, drugs leading to microcirculatory and mitochondrial function improvement might represent a therapeutic target. 

Fibrates were found to constitute promising candidates for positively influencing the course of sepsis. Their application showed, next to their primary effect, the ability to enable the treatment of dyslipidemia [7], an improved outcome in experimental bacterial and abdominal sepsis [8,9], and a positive effect on the severity of sepsis in a clinical study [10]. Fibrates are activators of intracellular peroxisome proliferator-activated receptor alpha (PPARα), expressed in various organs, including the liver and intestine [11,12]. Via PPARα, fibrates mainly influence the transcription of genes involved in mitochondrial β-oxygenation [13,14]. One potential mechanism behind fibrates’ beneficial impact on sepsis is their positive impact on mitochondrial function [13,14,15]. However, fibrates are also controversially discussed regarding their negative effects on healthy mitochondria, displaying a more profound need for research in the specific context of drug influences on mitochondria in the septic state [15]. But next to cell metabolism, fibrates act as immunomodulating agents, potentially positively influencing immunological processes during sepsis. So far, fibrates have been described to be able to enhance neutrophil recruitment [9] and decrease pro-inflammatory cytokines [16] and levels of inducible nitric oxide (NO) synthase [17]. This increase in NO synthase suggests an impact of fibrates on microcirculation, which has not been investigated in detail so far.

Therefore, the aim of this study is to determine if gemfibrozil, one agent of the group of fibrates, can impact abdominal sepsis positively, focusing in detail on aberrations of microcirculation in the colon and liver. Moreover, as stated above, since the effect of gemfibrozil on mitochondria remains a subject of discussion, the impact of gemfibrozil on mitochondrial function and lipid peroxidation levels as an indicator of oxidative stress was also monitored. Further, the here-studied effects of gemfibrozil were evaluated regarding their PPARα dependency by using the PPARα inhibitor GW6471. 

## 2. Results

### 2.1. Vital Parameters and Organ Damage Parameters

During the intervention, vital parameters were registered continuously (Table 1 and Table 2). Throughout all the sham groups, slight decreases in MAP, HR, and lactate levels were visible, but all the measurements remained in physiological ranges. Organ damage parameters were determined via EDTA blood samples; the results are displayed in Table 3. Gemfibrozil, GW 6471, or the combination of both drugs did not significantly impact levels of creatinine, urea, aspartate aminotransferase (AST), and alanine aminotransferase (ALT) in the sham animals.

The vital parameters of the CASP-operated groups are shown in Table 2. Overall, and similar to the sham-operated group, MAP and HR exhibited a slight downward trend across all groups. Still, they remained within the physiological range for both frequency and pressure (Table 2). Further, comparable to the sham animals, in the CASP animals, no significant impacts of gemfibrozil on renal and liver parameters were observed (Table 4).

### 2.2. Effect of Gemfibrozil on Microvascular Oxygenation in the Colon and Liver

The untreated sham-operated animals showed no significant changes in µHbO_2_ of the colon. After 60 min, gemfibrozil (30 mg/kg BW (−6.7 ± 2.7% *)) and gemfibrozil (100 mg/kg BW/GW6471 (−6.9 ± 8.4% *)) led to a significant decrease in µHbO_2_ levels. After 90 min, all intervention groups but the sham control group showed a substantial reduction in µHbO_2_ levels in the colon (gemfibrozil, 30 mg/kg BW: −8.1 ± 6.1% *; gemfibrozil, 100 mg/kg BW: −5.2 ± 7.4% *; gemfibrozil, 30 mg/kg BW/GW 6471: −4.5 ± 8.9% *; gemfibrozil, 100 mg/kg BW/GW 6471: −8.9 ± 8.9% *; GW 6471: −5.1 ± 6.4% * (Figure 1a). In the liver, there were no significant changes in microcirculatory parameters compared to the baseline in any of the groups for the sham-operated animals (Figure 1b). Nonetheless, in the sham animals, gemfibrozil (30 mg/kg BW) led to significantly improved levels (6.6 ± 11% *) compared to higher gemfibrozil dosages (−6.8 ± 10.7%) or GW 6471 (−8.8 ± 10.3%) after 60 min of the intervention. After 90 min, the levels of the gemfibrozil-treated animals (30 mg/kg BW) remained at a significantly higher µHbO_2_-level (5.3 ± 6.6% *) compared with the higher gemfibrozil dosage of 100 mg/kg BW (−7.2 ± 15.3%) (Figure 1b).

In contrast, the colonic parenchyma of the untreated CASP animals showed a significant decrease in µHbO_2_ 90 min after the laparotomy (−6.5 ± 7.0% *) (Figure 2a). Furthermore, the CASP animals treated with GW 6471 exhibited a significant decrease in µHbO_2_ after only 60 min (−4.6 ± 5.0% *) (Figure 2a). In contrast, treatment with 30 mg/kg BW of gemfibrozil (−2.4 ± 7.1%) or 100 mg/kg BW (−3.5 ± 4.2%) of gemfibrozil, either alone or in combination with GW 6471 (30 mg/kg BW + GW 6471: −0.6 ± 9.2%; 100 mg/kg BW + GW 6471: −3.1 ± 5.7%), prevented any significant decline in µHbO_2_ after 90 min (Figure 2a). Similar to the colon tissue, the oxygenation of the liver was significantly impaired in the CASP control animals 90 min after the laparotomy (−11.8 ± 4.3% *) (Figure 2b). Interestingly, in line with the findings for colon tissue, the gemfibrozil-treated animals did not exhibit any impairments in the hepatic microcirculation (gemfibrozil, 30 mg/kg BW: −0.22 ± 9.1; gemfibrozil, 100 mg/kg BW: −4.8 ± 11.9%; gemfibrozil, 30 mg/kg BW + GW 6471: −3.76 ± 9.54%; gemfibrozil, 100 mg/kg BW + GW 6471: 0.42 ± 8.87%) after 90 min. (Figure 2b). In contrast to the colon, the livers of the CASP animals treated with GW 6471 did not exhibit a decrease in µHbO_2_ (−1 ± 11.3%) (Figure 2). 

### 2.3. Mitochondrial Respiration and ATP Content

Mitochondrial function was evaluated by calculating the ADP/O ratio and the respiratory control index (RCI) for complexes I and II. In the sham animals, no significant changes in colonic mitochondrial function were observed in either complex I or complex II after gemfibrozil/GW 6471 treatment, as shown in Figure 3a–d. Analyses of the liver tissue also revealed no significant impacts of gemfibrozil on mitochondrial function (Figure 3e–h). Furthermore, in the sham animals, there were no significant changes in ATP concentrations in colon and liver tissue following gemfibrozil treatment, as depicted in Figure 4.

Parallel to the sham analyses, mitochondrial respiration and ATP content were analysed in the CASP animals. Calculations of the ADP/O ratio and the RCI did not reveal any significant changes in mitochondrial function in the colon or liver induced by the treatment with gemfibrozil in the CASP-operated animals, as shown in Figure 5. Moreover, there were no differences observed in the ATP concentrations in the colon or the liver tissue among the treatment groups (Figure 6).

### 2.4. Lipid Peroxidation Levels as an Indicator of Oxidative Stress

MDA assays were performed on the sham and CASP animals to evaluate the impact of gemfibrozil and PPARα inhibitors on lipid peroxidation levels, serving as one of the indicators of oxidative stress in colon and liver tissue. Neither gemfibrozil (30 mg/kg BW and 100 mg/kg BW), GW 6471, nor the combination of both substances significantly changed MDA levels in either organ, as displayed in Figure 7.

## 3. Discussion

Over the last two decades, several studies have reported a beneficial effect of lipid-lowering drugs on sepsis. In particular, the group of statins has been thoroughly analysed, and a beneficial effect on the 30-day survival of septic patients has been repeatedly demonstrated [18,19]. Interestingly, this beneficial effect varies among the individual statins, possibly depending on substance-specific pleiotropic immunomodulatory effects rather than their lipid-lowering effect [18]. Next to the direct antibacterial effects of statins, some mechanisms of statin-induced immunomodulation seem to be PPARα-dependent, as shown in vitro but also in vivo [20,21,22]. Regarding sepsis, interestingly, the beneficial potential of fibrates has been far less examined than statins, although they act similarly to statins via PPARα activation. Gemfibrozil has already been shown to attenuate experimental abdominal sepsis by reducing proinflammatory cytokine levels and organ damage parameters [8]. Whether these effects depend, on the one hand, on changes in microcirculation or, on the other hand, mechanistically on fibrate-induced PPARα activation has not been examined so far. In this paper, we aim to provide a comprehensive understanding of gemfibrozil’s action on abdominal sepsis and its dependence on PPARα by analysing microcirculation, mitochondrial function, and lipid peroxidation as an indicator of oxidative stress during abdominal sepsis under two dosages of gemfibrozil and in conjunction with PPARα inhibitor GW 6471. The following four main conclusions can be drawn by summarizing the results mentioned above.

In non-septic subjects, gemfibrozil treatment had an organ-dependent effect on microvascular oxygenation, displayed by stable µHbO_2_ measurements in the liver but a slight reduction in the colon.Gemfibrozil treatment prevented colonic and hepatic microvascular oxygenation aberrances occurring under septic conditions.Gemfibrozil seems to affect the microcirculation PPARα-independently.Gemfibrozil treatment did not affect mitochondrial function and lipid peroxidation levels, which serve as an indicator of oxidative stress.

To induce abdominal sepsis, the well-established CASP model was chosen [23,24]. The CASP model appears to be more advantageous than the coecum ligation and puncture (CLP) sepsis model for evaluating new treatment approaches for diffuse peritonitis and subsequent sepsis [25]. Twenty-four hours after CASP operation, evident signs of peritonitis were visible (oedema, haemorrhage, and fibrin coatings), but vital parameters remained in physiological ranges during the intervention without the need for circulation resuscitation. Previous studies conducted by our working group using the CASP model confirmed significant increases in proinflammatory cytokines such as tumour necrosis factor α, interleukin-6, and anti-inflammatory cytokine Il-10 at this time point [24]. This physiological setup aligns with that presented in Lustig et al. (2007), who initially adjusted the CASP model to rats and described circulatory changes in CASP animals in depth. Using a 16 G stent, they reported a mortality rate of 71% 48 h after CASP operation but stable vital parameters until one hour before death [23].

Here, gemfibrozil was administered in two dosages. Gemfibrozil at 100 mg/kg is the most commonly used dosage in rodents and has already been shown to decrease proinflammatory cytokines during experimental sepsis [8]. Moreover, gemfibrozil (100 mg/kg) showed antioxidative effects in diabetic rats when administered over two weeks [25]. In patients, the standard intake of gemfibrozil is 1200 mg/d per os, equivalent to ~17 mg/kg in terms of nearly 100% oral bioavailability. To investigate a dosage closer to the clinical use, we also administered 30 mg/kg, which lies slightly above the clinical dosage but was already shown to induce PPARα upregulation in a rodent model [26]. For PPARα antagonism, the PPARα-selective inhibitor GW 6471 was used [27]. GW 6471 mainly acts by blocking the active conformation of the AF-2 helix, leading to a modified pocket that enables the binding of co-repressor motifs [27]. The PPARα antagonist GW 6471 was injected at a 1 mg/kg dosage, which is the routinely used dosage to achieve significant PPARα antagonism-mediated effects [12,28,29].

The first result drawn from the microcirculatory records is the differential influence of gemfibrozil treatment depending on the state of the animal and, under non-septic conditions, also on the organ. Interestingly, the gemfibrozil-treated sham animals showed significantly decreased colon microcirculatory parameters during the intervention, reflected by significant reductions in µHbO_2_. In the gemfibrozil-treated sham animals, we also saw a slight decrease in MAP, which implies statistical rather than clinical relevance. Moreover, all MAP parameters as well as lactate levels remained in the physiological range. Therefore, the changes in the microcirculatory parameters were not caused by the macrocirculatory disturbances. There are some reports about the effects of fibrates on the microcirculation under pathological conditions like diabetes, metabolic syndrome, or atherosclerosis, but no studies were performed under almost physiological conditions like those after a sham operation. Due to these differing experimental setups, direct comparisons with other studies can only be made to a limited extent. Therefore, we can only speculate about the possible mechanisms behind our results. The observed reduction in µHbO_2_ in the groups treated with gemfibrozil and PPARα inhibitor GW6471 suggests a PPARα-independent mechanism of action. PPARα-independent pathways of gemfibrozil have already been studied, but they have been described in less detail [30]. One of these PPARα-independent pathways described is the activation of phosphatidylinositol-3 kinase (PI3K) by gemfibrozil [17]. PI3K isoforms are known targets of vasoconstrictors like angiotensin and endothelin-1 and mediate vasoconstriction via enhancing calcium currents [31,32,33,34], partly explaining the vasoactive potential and therefore µHbO_2_ alterations induced by gemfibrozil in the sham animals in this study. Interestingly, hepatic microcirculation was not observed to be affected by gemfibrozil treatment in the sham animals, demonstrating a gemfibrozil-induced organ-dependent circulatory redistribution. Concordantly, in the liver, activation of phosphatidylinositol-3 kinase/protein kinase b (PI3K/Akt) pathway has already been described to attenuate liver ischemia/reperfusion damage via activating endothelial nitric oxide synthase (eNOS) and increasing hepatic NO levels [35,36]. Gemfibrozil’s activation of PI-3 kinase might explain, therefore, impairments as well as improvements in local microcirculation. Whether this assumed pathway mediates gemfibrozil-induced microcirculatory changes remains speculative and is the subject of future investigations.

Regarding sepsis, microcirculation aberrations have already been identified as an independent indicator of a worse outcome [37]. To the best of our knowledge, we evaluated the effect of gemfibrozil on microcirculation in abdominal sepsis for the first time. There are studies available about the positive effects of fibrates on microcirculatory variables under pathological conditions like non-alcoholic fatty liver disease [38], metabolic syndrome [39], or diabetes [40], discussing different mechanisms underlying the improvement of microcirculation. Haak et al. (1998) observed lower fibrinogen levels in fenofibrate-treated hyperlipidaemic patients that led to reduced plasma velocity [41]. Kondo et al. (2010) registered significant hepatic microvascular patency and tissue oxygenation improvements in rats with non-alcoholic fatty liver disease treated with fenofibrate [38], while Harmer et al. (2015) reported a significant improvement in arterial endothelial function after four months of fenofibrate treatment [40]. 

In our study, the gemfibrozil-induced microcirculation improvements were not inhibited via PPARα antagonist; consequently, a solely PPARα-mediated mechanism appears unlikely. 

Nevertheless, PPARα may be partly involved in microcirculation regulation, as the PPARα-antagonist-treated CASP animals displayed ameliorated microcirculation in the liver parenchyma while also exhibiting microcirculatory decrease of the colon. One explanatory approach to these diverging findings may be organ-differential PPARα-expression. Standage et al. (2012) found lowered expression levels of PPARα in patients diagnosed with septic shock [42]. Regarding differential organ expression, Van Wyngene et al. (2020) found a downregulation of PPARα in hepatic tissue after inducing sepsis via caecal ligation and puncture, which correlated with sepsis severity [43]. We did not assess PPARα-expression in the colon and liver in this study, so this assumption remains speculative. To interpret the observed results according to these clinical findings, the organ-dependent expression of PPARα in abdominal sepsis should be investigated in future studies, especially as PPARα expression levels have already been shown to correlate with survival among septic patients [42].

The here-observed gemfibrozil-induced ameliorated microcirculation in the liver and colon under septic conditions is quite similar to results reported after pravastatin treatment. For pravastatin, we showed, most recently, a similar PPARα-independent improvement in µHbO_2_ in the intestine and liver [44]. Mechanistically, statins have been described as being able to increase eNOS expression, resulting in vasodilatation and modulated inflammatory responses, as well as increased tight junction density via the Caveolin-1/eNOS pathway [45,46]. Fibrates, in general, have also already been proven to increase eNOS expression in bovine aortic endothelial cells as well as human vascular endothelial cells [47,48], but data regarding gemfibrozil are still lacking. The potential activation of the eNOS pathway by gemfibrozil under septic conditions could be a possible explanation for the different routes of action of this drug under septic and non-septic conditions. Regarding the results of the hepatic microcirculatory analysis, it is striking that GW 6471 also prevents microcirculatory impairment. As the previous dataset did not display this preventive effect, another approach is required to re-examine this finding [44]. 

Another aim of this study was to determine gemfibrozil’s effects on mitochondrial function and the level of lipid peroxidation, serving as an indicator for oxidative stress. Preliminary results obtained in organ homogenates from healthy rats showed an organ- and dose-dependent effect of gemfibrozil on mitochondrial function. After gemfibrozil treatment with dosages similar to those used here in vivo, declined mitochondrial function in liver homogenates was detected, interestingly, together with improved mitochondrial function in the colon tissue [49]. Nadanaciva et al. also reported mitochondrial impairment (complex I) via gemfibrozil in isolated rat liver mitochondria [15]. Here, in contrast, gemfibrozil did not affect hepatic and colonic mitochondrial function, neither in the sham nor septic animals. Taking the underlying approaches, which display variations in methodology (ex vivo/in vivo) and underlying pathology (non-septic/abdominal sepsis), as a basis, divergent results are conceivable but still noteworthy. Consistent with the unchanged mitochondrial function, no changes in MDA levels, a marker of lipid peroxidation, were observed either. In the heart, PPARα-mediated improvement of mitochondrial function and ROS production in cardiomyocytes was already observed [50]. However, the lack of effects of gemfibrozil on mitochondrial function in our study prohibits assumptions about PPARα-mediated mitochondrial implications. 

The limitations of this study are mainly seen in the restricted transferability of the results based on animal models. As a rodent model was used here to obtain a deeper insight into gemfibrozil’s effects on microcirculation and mitochondrial function and the potential mechanisms behind them, the results should be translated into the clinical context with reasonable care. Further, statements about organ damage resulting from the observed decreased microcirculation parameters remain speculative, as no predefined levels of µHbO_2_ decreases causing cell death and resulting in organ damage are available. Moreover, in this study, the expression of PPARα in the intestine and liver was presumed based on the literature [11,12], but it was not assessed.

## 4. Materials and Methods

### 4.1. Animals

The study was approved by the local animal welfare committee (Landesamt für Natur, Umwelt und Verbraucherschutz, Recklinghausen, Germany, AZ. 84-02.04.2015.A398) and conducted according to the ARRIVE guidelines. Male Wistar rats were kept in standardized conditions (food and water provided ad libitum, 12 h day/night cycle, 20–22 °C). A total of 120 animals were randomly assigned to one of the twelve treatment groups displayed in Table 5 (n = 10). The temporal sequence of the experiments is shown in Figure 8.

### 4.2. CASP Model

Sepsis was induced using the colon ascendens stent peritonitis (CASP) model, as previously described [51]. Briefly, animals were administered buprenorphine (0.05 mg/kg body weight (BW) subcutaneously (s.c.)) followed by sevoflurane inhalation. A laparotomy was performed, and two 16 G stents were placed in the caecum. Sham animals were subjected to a similar operation, but 16 G stents were fixed only on the surface of the colon without mucosal penetration. After repositioning the caecum, 5 mL of NaCl 0.9% solution for volume restitution was applied intraperitoneally (i.p.), the laparotomy was closed, and anaesthesia was stopped. Every twelve hours, animals received analgesia (buprenorphine 0.05 mg/kg BW s.c.) and 1 mL of NaCl 0.9% s.c.

### 4.3. Treatment

Gemfibrozil or vehicle solution was applied intraperitoneally (i.p.) 24 h and directly before the CASP or sham operations. Two dosages of gemfibrozil (30 mg and 100 mg/kg BW) were chosen. The carrier substance dimethyl sulfoxide (DMSO) 50% served as control. Further, animals received PPARα inhibitor GW 6471 (1 mg/kg BW i.p.), which is the commonly used dosage for sufficiently inhibiting PPARα activation. DMSO 5% served as a vehicle. GW 6471 or DMSO was administered 0.5 h before the gemfibrozil injection [12,52,53].

### 4.4. Intervention

To enable re-laparotomy, animals were anaesthetized using buprenorphine (0.05 mg/kg BW s.c) and pentobarbital sodium (60 mg/kg BW i.p.). The absence of any movement and interdigital reflexes proved adequate anaesthesia depth. Animals were administered 3 mg/kg BW of pancuronium and subjected to a tracheotomy following arterial and central vein cannulation. Animals were ventilated using a gas mixture of 30% oxygen and 70% nitrogen. Blood pressure was recorded continuously. Every 30 min, blood gas analyses were performed. After re-laparotomy was completed, two O2C sensors (O2C LW 2222, Lea Medizintechnik GmbH, Gießen, Germany) were placed on colon ascendens and liver parenchyma. During the operation, animals received continuous volume substitution (Ringer’s-lactate-solution 4 mL/kg BW/h). After the measurement period of 90 min had elapsed, animals were euthanized via exsanguination. Obtained blood was stored in EDTA vials. Liver and colon parenchyma were harvested for further measurements described below.

### 4.5. Microcirculation Evaluation

As previously described, the microcirculatory variable µHbO_2_ was assessed using reflection spectroscopy (O2C LW 2222, Lea Medizintechnik GmbH, Gießen, Germany) [54]. In short, white light with a wavelength of 450–1000 nm was emitted into the tissue, where it was either absorbed by hemoglobin or scattered by cellular components [55,56]. The reflected light, therefore, differs from the initially emitted white light in terms of intensity and wavelength spectrum. After the registration of the reflected light by the emitting probe (Flat Probe LFX-2, LEA Medizintechnik GmbH, Gießen, Germany), the main spectrum was analysed by an integrated software, and the averaged microvascular oxygen saturation was calculated. Since blood is not homogeneously distributed within the microcirculation, the signal from venous capacitance vessels quantitatively dominates, and the registered spectrum is shifted to the venous region. In consequence, µHbO_2_ mainly indicates postcapillary oxygen saturation as a measure of regional oxygen reserve [57]. In the experiments, simultaneous measurement was used to determine µHbO_2_ of colon and liver at the same time. Measures were taken every two seconds and summarized as means of 5 min periods. The reported µHbO_2_ levels represent the means of the last 5 min of each 30 min interval under steady-state conditions.

### 4.6. Mitochondrial Respiratory Rates

For homogenization, freshly isolated colon tissue was placed in isolation buffer (200 mM of mannitol, 50 mM of sucrose; 5 mM of KH_2_PO_2_; 5 mM of 3-(N-morpholino)-propane sulfonic acid (3-MOPS); 0.1% bovine serum albumin BSA, 1 mM of ethylene glycol-bis-(beta-aminoethyl ether)-tetraacetic acid (EGTA), pH 7.15), faeces was washed off, and incubated with trypsin (0.05%) for 5 min. After being transferred into a 4 °C cold isolation buffer containing an additional 2% BSA and cOmplete™ Protease Inhibitor cocktail (Roche Life Science, Mannheim, Germany), tissue was shredded and homogenised (Potter-Elvehjem, 5×, 2000 rpm). Similarly, freshly isolated liver tissue was placed in an isolation buffer and shredded. Buffer was replaced, and tissue was homogenised (Potter-Elvehjem, 5×, 2000 rpm). Protein concentrations were determined according to the Lowry method [56]. Samples were kept on ice during all steps of the sample preparation procedure.

As previously described, mitochondrial respiratory rates were analysed using a Clark-type electrode (model 782, Strathkelvin Instruments, Glasgow, Scotland) [49]. Homogenates were adjusted to equal protein content by adding respiratory medium (KCl 130 mM, K_2_HPO_4_ 5 mM, MOPS 20 mM, EGTA 2.5 mM, Na_4_P_2_O_7_ 1 µM, and BSA 0.1% for liver and BSA 2% for colon, pH 7.15). Colon tissue homogenates (6 mg /mL protein content) and liver tissue homogenates (4 mg/mL protein content) were further examined at 30 °C, ensuring uniform oxygen solubility across all samples. The addition of complex I the substrates glutamate (2.5 mM) and malate (2.5 mM), or complex II substrate succinate (10 mM for liver, 5 mM for colon analyses) with an inhibitor of complex I (Rotenone 0.5 µM, Sigma-Aldrich Corporation, St. Louis, MO, USA) enabled the mitochondrial state 2 respiration rate measurement [49]. Accordingly, the addition of adenosine diphosphate (250 μM ADP for liver, 50 μM ADP for colon, Sigma-Aldrich Corporation, St. Louis, MO, USA) enabled the measurement of maximal coupled mitochondrial respiration, defined as state 3. The respiratory control index (RCI) was calculated by relating state 3 results to state 2 results, reflecting the coupling between oxidative phosphorylation and the electron transport system [58]. By relating the ADP amount added in state 3 to the consumed oxygen, the oxidative phosphorylation (OXPHOS) efficacy as an ADP/O ratio was calculated.

### 4.7. Malondialdehyde-Assay

For malondialdehyde assay, liver and colon tissue were thawed (originally being −80 °C) and homogenised in potassium chloride (1.15%, 500 µL). Half of the sample volume was added to 1.5 mL of phosphoric acid (1%) and 0.5 mL of thiobarbituric acid (0.6%). Samples were heated to 95 °C for 45 min and subsequently stored on ice. Samples were centrifuged again after the addition of 2 mL of butanol. Then, a 200 µL sample volume was mixed with an equal volume of potassium chloride. Absorbance was determined via photospectrometry at 535 and 520 nm. The malondialdehyde concentration was calculated by normalizing the MDA standard against protein content determined via the Lowry method. MDA content was expressed as nmol MDA/mg protein.

### 4.8. ATP-Measurement

Colon or liver samples (each 50 mg) frozen in liquid nitrogen were homogenised in Tris-HCl buffer (20 mM Tris, 135 mM KCl, pH 7.4). A 50 µL homogenised sample was added to 450 µL of buffer (100 mM Tris; 4 mM Ethylenediaminetetraacetic acid (EDTA), pH 7.75) and incubated for 2 min (100 °C), followed by centrifugation at 1000 g for 2 min. For ATP content determination, the ATP Bioluminescence Assay kit CLS II (Roche, Basel, Switzerland) was used.

### 4.9. Plasma Analyses

EDTA Blood samples were centrifuged (4 °C, 4000× *g*, 10 min), and the obtained plasma was stored at −80 °C. Samples were further analysed in the Central Institute of Clinical Chemistry and Laboratory Medicine of the University Hospital Duesseldorf, Germany, determining the organ damage parameters of the liver (alanine aminotransferase (ALT), aspartate aminotransferase (AST)), and of the kidneys (creatinine and urea). Parameters were analysed using UV tests according to a standardised method for Roche MODULAR analysers.

### 4.10. Statistics

Results were analysed using Graph Pad Prism v6.01 (Graph Pad Software, Inc., Boston, MA, USA). Microcirculation measurements were analysed using two-way ANOVA for repeated measures, followed by the Dunnett post hoc test for alterations vs. baseline and Tukey post hoc for alterations within the sham and CASP groups. Data are presented as means + standard deviation and show the deviation from the baseline. Kruskal–Wallis Test, together with Dunn’s post hoc test, was used for all other data. Comparisons were drawn between the six treatment groups within the sham or CASP cohort. Data are shown as min/median/max. *p* < 0.05 was considered significant.

## 5. Conclusions

In conclusion, the effect of gemfibrozil depends on the underlying condition and differs between septic and non-septic states. Gemfibrozil mediated positive effects on colon and liver microcirculation in a CASP model of abdominal sepsis in rats. These results imply that fibrates can reveal a protective pathway that should be paid attention to in future sepsis treatment. Further investigations are needed to clarify the fibrates’ mechanisms of action and the role of PPARα in their pleiotropic effects. 

## Figures and Tables

**Figure 1 ijms-25-00262-f001:**
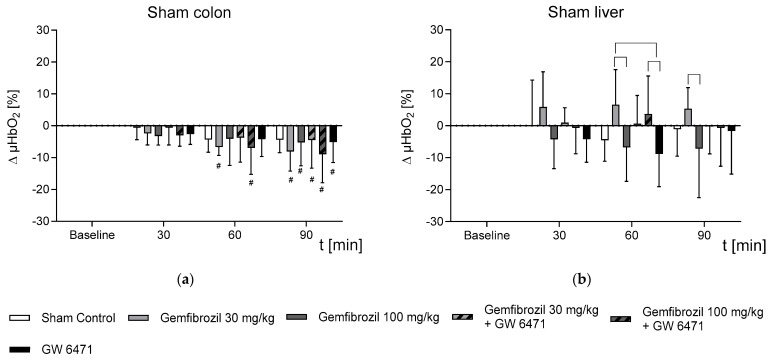
Temporal changes in postcapillary oxygenation (µHbO_2_) of the (**a**) colon and (**b**) liver in sham animals during 90 min intervention (laparotomy). The cohorts received either nothing (control), gemfibrozil (30 mg/kg BW or 100 mg/kg BW), PPARα inhibitor (GW 6471), or a combination of both. Data are presented as the difference from baseline in absolute percentage points and shown as means ± SD; # *p* < 0.05 vs. baseline was considered significant; ┌─┐ *p* < 0.05 vs treatment groups was considered significant.

**Figure 2 ijms-25-00262-f002:**
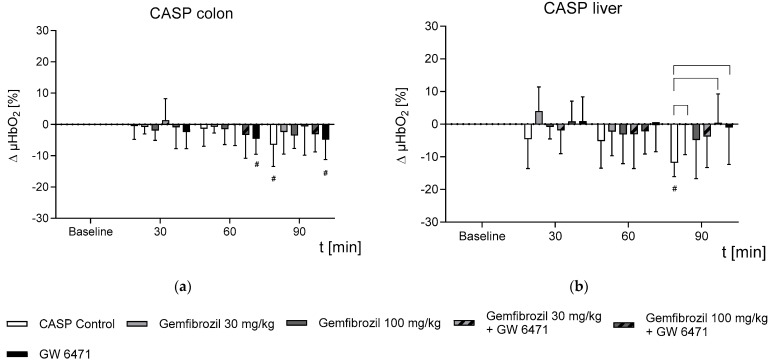
Temporal changes in postcapillary oxygenation (µHbO_2_) of the liver in CASP-operated animals during 90 min intervention (laparotomy). The cohorts received either nothing (control), gemfibrozil (30 mg/kg BW or 100 mg/kg BW), PPARα inhibitor (GW 6471, 1 mg/kg BW), or a combination of both drugs. Data are presented as the difference from baseline in absolute percentage points and shown as means ± SD; # *p* < 0.05 vs. baseline and ┌─┐ *p* < 0.05 vs. treatment groups were considered significant.

**Figure 3 ijms-25-00262-f003:**
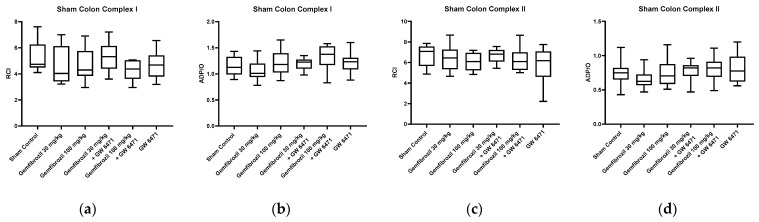
Colonic and hepatic mitochondrial respiration in sham animals. Analysis of colonic and hepatic mitochondrial respiration for complexes I (**a**,**b**,**e**,**f**) and II (**c**,**d**,**g**,**h**) after treatment with gemfibrozil (30 or 100 mg/kg BW), GW6471 (1 mg/kg BW), and the combination of both: RCI (state 3/state 2) (**a**,**c**,**e**,**g**), and ADP/O = ADP added/oxygen consumed in state 3 (**b**,**d**,**f**,**h**). Data are presented as medians with whiskers from min to max; n = 10.

**Figure 4 ijms-25-00262-f004:**
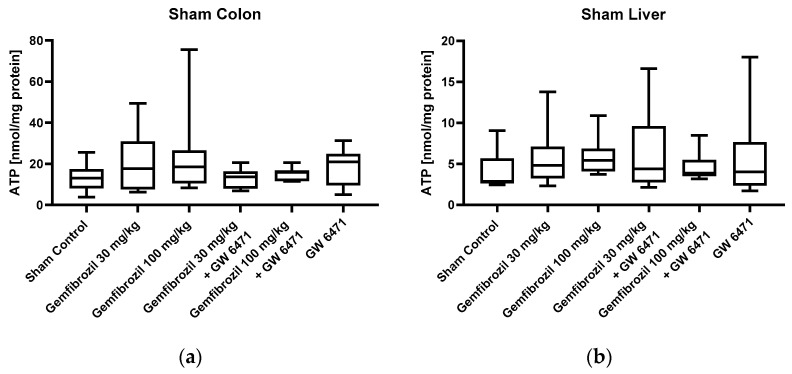
ATP levels of the sham-operated animals in colon (**a**) and liver (**b**). The cohorts received either nothing (control), gemfibrozil (30 or 100 mg/kg BW), PPARα inhibitor (GW 6471 1 mg/kg BW), or a combination of both. The ATP Bioluminescence Assay kit CLS II was used to measure ATP levels. Data are displayed as medians with whiskers from min to max; n = 8–12.

**Figure 5 ijms-25-00262-f005:**
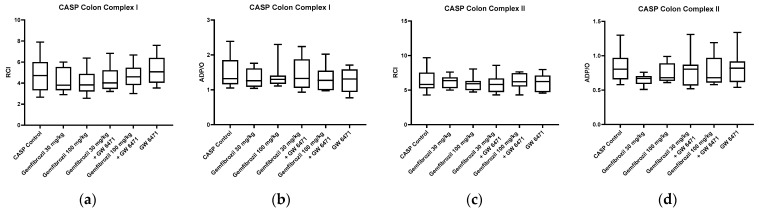
Colonic and hepatic mitochondrial respiration in CASP animals. Analysis of colonic and hepatic mitochondrial respiration for complexes I (**a**,**b**,**e**,**f**) and II (**c**,**d**,**g**,**h**) after treatment with gemfibrozil (30 or 100 mg/kg BW), GW6471 (1 mg/kg BW), and the combination of both: RCI (state 3/state 2) (**a**,**c**,**e**,**g**), and ADP/O = ADP added/oxygen consumed in state 3 (**b**,**d**,**f**,**h**). Data are presented as medians with whiskers from min to max; n = 10.

**Figure 6 ijms-25-00262-f006:**
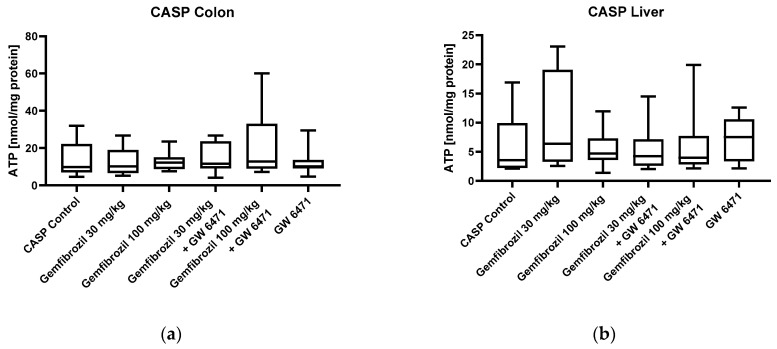
ATP levels of the CASP-operated animals in colon (**a**) and liver (**b**). The cohorts received either nothing (control), gemfibrozil (30 mg/kg BW or 100 mg/kg BW), PPARα inhibitor (GW 6471 1mg/kg BW), or a combination of both. The ATP Bioluminescence Assay kit CLS II was used to measure ATP levels. Data are displayed as boxplots and medians with whiskers from min to max; n = 10–12.

**Figure 7 ijms-25-00262-f007:**
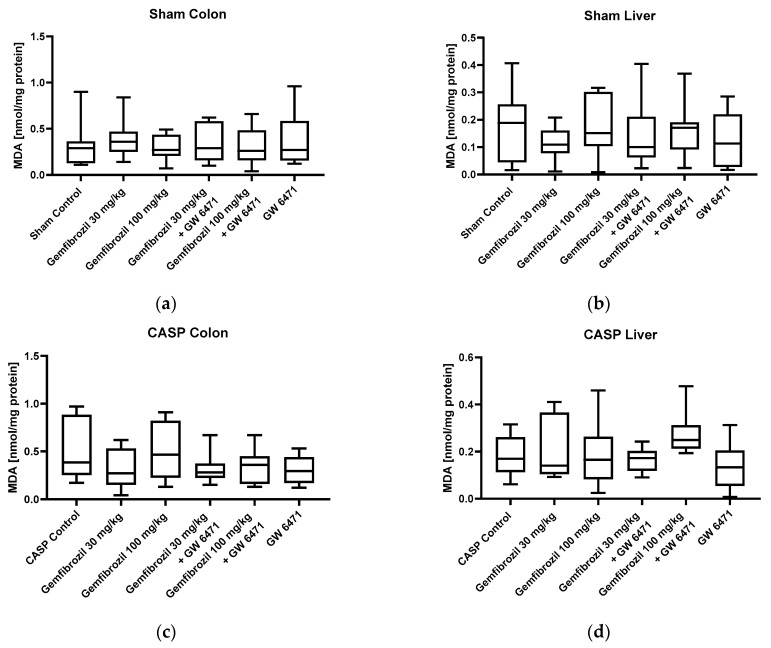
Lipid peroxidation levels. Analyses of MDA levels of sham-operated animals (**a**,**b**) and CASP-operated animals (**b**,**d**) in colon (**a**,**c**) and liver (**b**,**d**) after treatment with gemfibrozil (30 or 100 mg/kg BW), GW 6471 (1 mg/kg BW), or the combination of both. Data are presented as medians with whiskers from min to max; n = 10.

**Figure 8 ijms-25-00262-f008:**
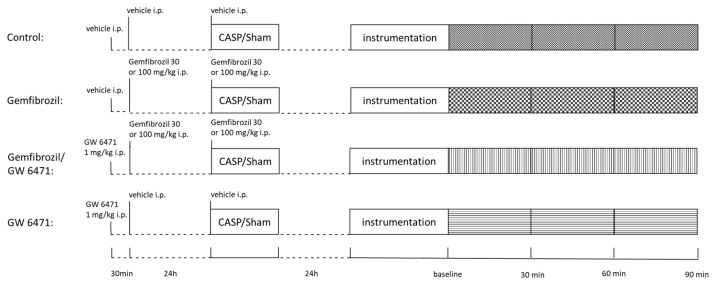
Temporal sequence of treatment/intervention. Gemfibrozil was injected twice, 24 h, and directly before CASP/sham surgery. GW 6471 was injected 30 min before the first gemfibrozil treatment. Twenty-four hours after CASP/sham surgery, intervention via re-laparotomy was performed, and parameters were registered for 90 min. After 90 min, exsanguination, organ removal, and mitochondrial analyses were performed.

**Table 1 ijms-25-00262-t001:** Vital parameters and lactate levels of sham-operated animals after pre-treatment with no substance (sham control), gemfibrozil (30 or 100 mg/kg BW), gemfibrozil in addition to PPARα inhibitor GW 6471, or GW 6471 (1 mg/kg BW) alone.

	Sham Control	Gemfibrozil 30 mg/kg BW	Gemfibrozil 100 mg/kg BW	Gemfibrozil 30 mg/kg BW + GW 6471	Gemfibrozil 100 mg/kg BW + GW 6471	GW 6471
Mean arterial pressure (MAP) [mmHg]						
Baseline	110 ± 22	106 ± 36	130 ± 19	124 ± 26	136 ± 10	127 ± 32
30 min	104 ± 24	97 ± 30	114 ± 31	108 ± 22	104 ± 34 *	113 ± 36
60 min	96 ± 27	86 ± 21 *	104 ± 37 *	93 ± 22 *	99 ± 26 *	102 ± 35 *
90 min	95 ± 32	84 ± 23 *	108 ± 37 *	91 ± 32 *	94 ± 34 *	105 ± 31 *
Heart rate (HR) [beats/min]						
Baseline	436 ± 58	485 ± 74	459 ± 35	481 ± 43	489 ± 58	467 ± 49
30 min	406 ± 58	474 ± 72	447 ± 40	450 ± 55 *	441 ± 45 *	444 ± 49 *
60 min	396 ± 51 *	445 ± 71 *	419 ± 40 *	430 ± 35 *	416 ± 39 *	396 ± 62 *
90 min	383 ± 63 *	429 ± 66 *^§^	430 ± 50	410 ± 53 *^§^	415 ± 43 *	380 ± 54 *^§^
Lactate [mmol/L]						
Baseline	1.7 ± 0.6	1.5 ± 0.4	1.2 ± 0.1	1.3 ± 0.6	1.3 ± 0.5	1.3 ± 0.70
30 min	1.2 ± 0.4 *	1.1 ± 0.3	1.2 ± 0.2	1.0 ± 0.1	1.4 ± 0.3	1.2 ± 0.67
60 min	1.1 ± 0.2 *	1.1 ± 0.4	1.2 ± 0.2	0.9 ± 0.4	1.3 ± 0.2	1.1 ± 0.33
90 min	1.1 ± 0.3 *	0.9 ± 0.2 *	0.9 ± 0.1	0.8 ± 0.3	1.3 ± 0.3	0.9 ± 0.28 *

Mean arterial blood pressure (MAP) and heart rate (HR) measures represent the last 5 min mean of every 30 min interval. Data are presented as means with standard deviation (SD) * *p* < 0.05 vs. baseline, § *p* < 0.05 vs. 30 min; n = 10.

**Table 2 ijms-25-00262-t002:** Vital parameters and lactate levels of CASP-operated animals after pre-treatment with no substance (CASP control), gemfibrozil (30 mg/kg or 100 mg/kg BW), gemfibrozil in addition to PPARα inhibitor GW 6471, or GW 6471 (1 mg/kg BW) alone.

	CASP Control	Gemfibrozil 30 mg/kg BW	Gemfibrozil 100 mg/kg BW	Gemfibrozil 30 mg/kg BW + GW 6471	Gemfibrozil 100 mg/kg BW + GW 6471	GW 6471
Mean Arterial Pressure (MAP) [mmHg]						
Baseline	105 ± 23	108 ± 18	108 ± 20	123 ± 26	109 ± 27	123 ± 26
30 min	93 ± 26	97 ± 16	107 ± 22	102 ± 21 *	102 ± 39	111 ± 36
60 min	85 ± 31 *	93 ± 21	98 ± 28	100 ± 24 *	103 ± 40	103 ± 29 *
90 min	90 ± 38	92 ± 21 *	99 ± 34	101 ± 25 *	97 ± 32	101 ± 27 *
Heart rate (HR) [beats/min]						
Baseline	477 ± 46	496 ± 40	484 ± 51	470 ± 75	478 ± 37	480 ± 64
30 min	432 ± 60 *	454 ± 57 *	461 ± 46	436 ± 65	441 ± 35 *	448 ± 62
60 min	420 ± 81 *	430 ± 54 *	443 ± 46 *	425 ± 66 *	431 ± 69 *	437 ± 56 *
90 min	417± 78 *	423 ± 62 *	427 ± 81 *	404 ± 56 *	440 ± 45 *	415 ± 48 *
			Lactate [mmol/L]			
Baseline	1.5 ± 0.5	1.9 ± 0.5	1.5 ± 0.6	1.4 ± 0.8	1.4 ± 0.6	1.4 ± 0.3
30 min	1.4 ± 0.9	1.6 ± 0.6	1.3 ± 0.5	1.2 ± 0.5	1.3 ± 0.6	1.3 ± 0.3
60 min	1.4 ± 0.6	1.3 ± 0.4 *	1.2 ± 0.4	0.9 ± 0.3 *	1.2 ± 0.5	1.5 ± 0.5
90 min	1.2 ± 0.5	1.0 ± 0.2 *^§^	1.0 ± 0.3 *	0.8 ± 0.2 *	1.2 ± 0.5	1.4 ± 0.3

Mean arterial blood pressure (MAP) and heart rate (HR) measurements represent the last 5 min mean of every 30 min interval. Data are presented as means with standard deviation (SD) * *p* < 0.05 vs. baseline, § *p* < 0.05 vs. 30 min; n = 10.

**Table 3 ijms-25-00262-t003:** Organ damage parameters of sham-operated animals after pre-treatment with no substance (sham control), gemfibrozil (30 or 100 mg/kg BW), gemfibrozil in addition to PPARα inhibitor GW 6471, or GW 6471 (1 mg/kg BW) alone.

	Sham Control	Gemfibrozil 30 mg/kg BW	Gemfibrozil 100 mg/kg BW	Gemfibrozil 30 mg/kg BW + GW 6471	Gemfibrozil 100 mg/kg BW + GW 6471	GW 6471
Creatinine (mg/dL)	0.35 ± 0.13	0.37 ± 0.1	0.46 ± 0.3	0.29 ± 0.1	0.37 ± 0.2	0.33 ± 0.1
Urea (mg/dL)	47.4 ± 11.7	50.6 ± 12.3	51.8 ± 14.0	44.7 ± 8.5	43.6 ± 13.1	47.8 ± 7.9
AST (U/L)	123.8 ± 74.2	200.5 ± 140.3	143.5 ± 60.7	117.8 ± 46.2	150.6 ± 109.1	103.3 ± 34.7
ALT (U/L)	55.6 ± 20.9	82.8 ± 56	52 ± 17.8	58.6 ± 23.6	80.8 ± 68.1	48.5 ± 14.1

Blood samples were taken 90 min after re-laparotomy and analysed for creatinine, urea, aspartate aminotransferase (AST), and alanine aminotransferase (ALT), serving as organ damage parameters. Data are presented as means with standard deviation (SD).

**Table 4 ijms-25-00262-t004:** Organ damage parameters of CASP-operated animals after pre-treatment with no substance (CASP-control), gemfibrozil (30 mg/kg BW or 100 mg/kg BW), gemfibrozil in addition to PPARα inhibitor GW 6471, or GW 6471 (1 mg/kg) alone.

	CASP-Control	Gemfibrozil 30 mg/kg BW	Gemfibrozil 100 mg/kg BW	Gemfibrozil 30 mg/kg BW + GW 6471	Gemfibrozil 100 mg/kg BW + GW 6471	GW 6471
Creatinine (mg/dL)	0.33 ± 0.1	0.32 ± 0.1	0.29 ± 0.1	0.33 ± 0.1	0.36 ± 0.1	0.35 ± 0.1
Urea (mg/dL)	45.5 ± 12.5	46.8 ± 11.4	43.9 ± 12.9	44.4 ± 11.2	46.1 ± 8.3	47.8 ± 10.8
AST (U/L)	149.7 ± 96.6	144.3 ± 36.6	175.5 ± 133.6	131.4 ± 40.1	204.3 ± 156	236.2 ± 307.9
ALT (U/L)	68.1 ± 51.1	56.4 ± 23.1	77.8 ± 71.5	57.6 ± 26.4	103.5 ± 91.6	141.6 ± 240.9

Blood samples were taken 90 min after re-laparotomy to assess organ damage levels and analysed for changes in creatinine, urea, aspartate aminotransferase (AST), and alanine aminotransferase (ALT). Data are presented as means with standard deviation (SD).

**Table 5 ijms-25-00262-t005:** Experimental groups.

Group 1	Sham + DMSO 50% (gemfibrozil carrier) and DMSO 5% (GW 6471 carrier)
Group 2	Sham + Gemfibrozil 100 mg/kg BW + DMSO 5%
Group 3	Sham + Gemfibrozil 30 mg/kg BW + DMSO 5%
Group 4	Sham + Gemfibrozil 100 mg/kg BW + GW 6471 1 mg/kg BW
Group 5	Sham + Gemfibrozil 30 mg/kg BW + GW 6471 1 mg/kg BW
Group 6	Sham + GW 6471 1 mg/kg BW + DMSO 50%
Group 7	CASP + DMSO 5% + DMSO 50%
Group 8	CASP + Gemfibrozil 100 mg/kg BW + DMSO 5%
Group 9	CASP + Gemfibrozil 30 mg/kg BW + DMSO 5%
Group 10	CASP + Gemfibrozil 100 mg/kg BW + GW 6471 1 mg/kg BW
Group 11	CASP + Gemfibrozil 30 mg/kg BW + GW 6471 1 mg/kg BW
Group 12	CASP + GW 6471 1 mg/kg BW + DMSO 50%

Sham and CASP animals were treated with dimethyl sulfoxide (DMSO) as a carrier substance, gemfibrozil at 30 or 100 mg/kg body weight (BW), gemfibrozil at 30 mg/kg or 100 mg/kg BW in combination with GW 6471 1 mg/kg BW, or GW 6471 alone. All drugs were administered intraperitoneally. Sham and CASP (colon ascendens stent peritonitis) animals were treated with dimethyl sulfoxide (DMSO) as a carrier substance, gemfibrozil at 30 or 100 mg/kg body weight (BW), gemfibrozil at 30 mg/kg BW or 100 mg/kg BW in combination with GW 6471 at 1 mg/kg BW, or GW 6471 alone. All drugs were administered intraperitoneally.

## Data Availability

The data presented in this study are available on request from the corresponding author.

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
