# Peer review of "Gemfibrozil Improves Microcirculatory Oxygenation of Colon and Liver without Affecting Mitochondrial Function in a Model of Abdominal Sepsis in Rats"

_ijms, 2023, doi:10.3390/ijms25010262_

Round 1
Reviewer 1 Report
Comments and Suggestions for Authors
The authors have shown that gemfibrozil improves microcirculatory oxygenation of colon while liver PPARα does not affect mitochondrial function in a model of abdominal sepsis in mice and rats. Many of the effects of gemfibrozil have been described in the literature. The drug is currently used in humans with excessive levels of lipids in the serum. The current report raises several concerns.
1. The authors have used mice and rats. The figures do not indicate if results are from mice or rats, and the fact that both were used makes it hard to dissect the results, requiring laborious searches of the text.
2. The authors have used a form of induction of sepsis in the colon employing a colonic puncture and manipulation of the colon that is different from the more standard cecal ligation and puncture (CLP). The fact that their form of induction of sepsis may have some different outcomes from the usual sepsis inductions needs to be commented on, comparing their method with the more standard form using CLP, which has been a common approach employed for nearly 50 years.
3. The take-away message from this report is that there seems to be fundamental differences in the responses in sepsis. I am not convinced that this is the case. The authors should consider the CLP strategy since this is the more standard model for the study of sepsis.
4. It might be instructive for the authors to evaluate how their interventions affect the more standard forms of ischemia-reperfusion (I/R) using the established forms involving the lungs, liver and kidney in which I/R injury is well established, to see to what extent such injury is affected by the interventions employing PPARα or gemfibrozil.
Reviewer 2 Report
Comments and Suggestions for Authors
Dear Editor/Authors,
Manuscript ID: ijms-2713046 entitled: "Gemfibrozil improves microcirculatory oxygenation of colon 2 and liver PPARα-independently without affecting mitochon- 3 drial function in a model of abdominal sepsis in rats" by authors: Kuebart et al., represent a study on the effects of gemfibrozil on microcirculatory variables, mitochondrial function and oxidative stress in the colon and liver under septic conditions, as well as its dependence on PPARα mediated mechanisms. The authors concluded that gemfibrozil influences the microcirculation depending on the underlying condition and prevents PPARα-independent sepsis-induced microcirculatory disturbances in the colon and liver. In non-septic animals, gemfibrozil affects the microcirculatory variables only in the colon.General comments:
Title
The title should be changed. It is to speculative beside authors did not measure PPARα expression.
Abstract
Written in a clear and understandable manner. Contains all elements necessary to understand the manuscript.
Introduction
The introduction is mostly well written, but the section on fibrates is in my opinion a bit too long.
Some parts of the introduction should be transferred to the discussion. In this way the basic concept of the manuscript would not be lost.
Results
The quality and presentation of the results are good. The text part is also good, but due to the large number of the results, it is a bit difficult to follow.
The drawback is that the authors did not state what the mortality of the rats was after the intervention. This is especially important for the group with induced sepsis.
Perhaps the tables for the sham and CASP animal groups should be combined (Tables 1 and 3 and Tables 2 and 4). Obviously, these are two separate experiments and one gets the impression that we have two different controls. I did not realize that the authors compared the parameters between sham and CASP groups of animals.
Subsection 2.4. MDA levels per se cannot be considered a measure of oxidative stress, but of lipid peroxidation. Oxidative stress is a broad term and includes enzymatic and low molecular weight components, MDA, PCO and oxidative DNA damage parameters. Should be rephrased throughout the text and use the term "lipid peroxidation level" instead of "oxidative stress level".
Discussion
The discussion is confusingly written. It does not follow the results, but relies on speculation about possible mechanisms. The authors should interpret their results. There is too much speculation about PPARα, but the expression of this parameter has not been determined at all. A complete revision of the discussion is needed.Lines 217-227 this part should be in M&M and not in the discussion
The biggest shortcoming of the study is that authors did not asses PPARα expression.
Materials and Methods
All animal experiments have been approved by the appropriate institution. The authors only stated the total number of animals in the experiment (120) in the abstract. In M&M, it is not stated, nor is the number of animals per group. Compliance with 3R guidelines is discutable. Authors used standard surgical techniques. The authors did not specify how the standard biochemical analyzes (ALT, AST, etc.) were determined. Appropriate statisticalal analysis for these type of analyses were performed.
Conclusions
The conclusions are too speculative.
All other comments are given in the text.
Conclusion of the Reviewer
The manuscript ijms-2713046 contains a large number of results, but also a large number of shortcomings mentioned in the previous comments. The discussion does not adequately explain the results obtained, but relies on the PPARα parameter, the expression of which was not determined at all in the experiments. Therefore, it is necessary to completely reorganize the entire text from the title to the conclusions. A large number of animals were used, which is understandable due to the high mortality due to sepsis. Therefore, this should be mentioned in the text. It is necessary for the authors to include the approval of the ethics committee for review by the reviewers. The results obtained are of scientific importance, but their interpretation is confusing. There are also a lot of technical and typographical errors. Therefore, in my opinion, the manuscript should be reconsider after major revision.
General conclusion: Reconsider after major revision.

Reviewer 3 Report
Comments and Suggestions for Authors
This is a well-conducted study. The experiments were well designed and performed. The methods are described correctly. The authors showed that gemfibrozil prevented the decrease in microcirculatory oxygenation in colon and liver associated to sepsis in rats. They used an inhibitor of PPARalpha (GW 6471) that allowed establishing that the observed effects are PPAR-independent. They observed that in control animals gemfibrozil damages microcirculatory oxygenation in colon but not in liver.
They showed that the data presents are nor related to alterations in mitochondrial function or oxidative damage.
Major:
Figures 1 and 2 (that show the key results) should be better explained and presented (see below). For example, each statement in the text should be supported by the % of changes expressed as numerical data and the corresponding significance, if any). All these statement should be consistent with all the statements about microcirculatory oxygenation in the rest of the manuscript including abstract and discussion). These changes are key support the conclusions of the article.
Figures 1 and 2. Why the SD is very high (mainly in liver data)?
Minor:
Line 25: MDA must be defined.
Line 50. ATP must be defined. Note: all abbreviations must be defined the first time used. Please avoid repetitions
Line 149: “respiratoíon”should be “respiration”
Line 309: “Wyngene” must be “Van Wyngene”
Line 356: “Table 84. 04.2015.A398)”???
Lines 443 and 444: The following phrase “The respiratory control index (RCI) was calculated by relating state 3 results to state 2 results…” must be supported by a reference.
Line 444: The definition of RCI is repeated. It had been previously defined in line 158.
Figures 1 and 2 (in which the key results of the study are presented) are very small and the symbols used are blurry. It is suggested to put that panel (a) above and the panel (b) in the bottom- In addition, the colors used to distinguish the different groups are similar. It may be better if contrasting colors are used instead.
Legends to figures. All the abbreviations used should be defined consisteltly.
Legends to figures 1, 2, and 4: “Data is” must be” “Data are”
Table 1: The following legend must be as a footnote: “Mean arterial blood pressure (MAP) and heart rate (HR) measures represent the last 5 min mean of every 30-minute interval. * p < 0.05 vs baseline, § p < 0.05 vs 30 min; n = 10.”
Table 2: The following legend must be as a footnote (below the table): “Blood samples were taken 90 minutes after re-laparotomy and 97 analyzed for creatinine, urea, aspartate aminotransferase (AST), and alanine aminotransferase 98 (ALT) serving as organ damage parameters.”
Table 3: The following legend must be as a footnote (below the table): “Mean arterial blood pressure (MAP) and heart rate (HR) measures represent the last 5 min mean of every 30-minute interval. * p < 0.05 vs baseline, § p < 0.05 vs 30 min; n = 10.”
Table 4: The following legend must be as a footnote (below the table): “Blood samples were taken 90 minutes after re-lap- 112 arotomy to assess organ damage levels and analyzed for changes in creatinine, urea, aspartate ami- 113 notransferase (AST), and alanine aminotransferase (ALT).”
Table 5: The following legend must be as a footnote (below the table): “Sham and CASP animals were treated with dimethyl sulfoxide (DMSO) as a carrier substance, gemfibrozil in 30 or 100 mg/kg body weight (BW), gemfibrozil in 30 mg/kg or 100 mg/kg body weight in combination with GW 6471 1 mg/kg body weight or GW 6471 alone. All drugs were administered intraperitoneally.” In addition CASP should be defined.
Round 2
Reviewer 1 Report
Comments and Suggestions for Authors
No additional comments to the authors.
Reviewer 2 Report
Comments and Suggestions for Authors@@
Dear Editor/Authors,
After a close reading of version V.2 of the manuscript IJMS-2713046, I have to admit that the authors were very wise in their approach to correcting their first version of the manuscript. They accepted all suggestions and gave logical explanations where they could not accept them. Considering the importance of sepsis as a global medical problem, the authors' great efforts to conduct a robust experiment, and the desire to publish the best manuscript possible, I believe that the manuscript IJMS-2713046 now is acceptable for publication in IJMS.
General conclusion: Acceptable for publication.
